# Biomimetic Grooved Ribbon Aerogel Inspired by the Structure of *Pinus sylvestris var. mongolica* Needles for Efficient Air Purification

**DOI:** 10.3390/polym17091234

**Published:** 2025-04-30

**Authors:** Bo Zhao, Zikun Huang, Mingze Han, Bernardo Predicala, Qiushi Wang, Yunhong Liang, Mo Li, Xin Liu, Jiangtao Qi, Li Guo

**Affiliations:** 1Key Laboratory of Bionic Engineering, Ministry of Education of China, Jilin University, Changchun 130022, China; bozhao21@mails.jlu.edu.cn (B.Z.); hanmz22@mails.jlu.edu.cn (M.H.); wangqs22@mails.jlu.edu.cn (Q.W.); liangyunhong@jlu.edu.cn (Y.L.); moli@jlu.edu.cn (M.L.); lx2017@jlu.edu.cn (X.L.); 2College of Biological and Agricultural Engineering, Jilin University, Changchun 130022, China; huangzk9922@mails.jlu.edu.cn; 3Jilin Provincial Key Laboratory of Smart Agricultural Equipment and Technology, Changchun 130022, China; 4Prairie Swine Centre (PSC), Saskatoon, SK S7H 5N9, Canada; bernardo.predicala@usask.ca

**Keywords:** electrospinning, grooved ribbon fiber, aerogel, air filtration, ammonia adsorption

## Abstract

Air pollutants, such as particulate matter (PM) and ammonia (NH_3_), generated by intensive animal farming pose considerable threats to human health, animal welfare, and ecological balance. Conventional materials are often ineffective at simultaneously removing multiple pollutants, maintaining a low pressure drop, and ensuring durability in heavily polluted environments. Inspired by the dust-retention properties of *Pinus sylvestris var. mongolica* (PS) needles, this study developed a biomimetic grooved ribbon fiber using electrospinning technology. These fibers were further assembled into a three-dimensional bioinspired aerogel structure through freeze-forming technology to achieve efficient dust capture. Additionally, the introduction of UiO-66-NH_2_ nanoparticles significantly enhanced the properties of the aerogels for NH_3_ adsorption. Among the various prepared aerogels (PG, UPG-5, UPG-10, UPG-15, and UPG-20), UPG-10 demonstrated the best performance, achieving a filtration efficiency of 99.24% with a pressure drop of 95 Pa. Notably, it exhibited a remarkable dust-holding capacity of 147 g/m^2^, and its NH_3_ adsorption capacity reached 99.89 cm^3^/g, surpassing PG aerogel by 31.46 cm^3^/g. Additionally, UPG-10 exhibited outstanding elasticity, maintaining over 80% of its original shape after 30 compression cycles. This biomimetic aerogel presents a promising solution for air purification, contributing to improved agricultural efficiency and environmental sustainability.

## 1. Introduction

Air pollutants emitted from intensive animal farming are significantly different from those commonly found in the atmosphere [1]. These pollutants originate from complex sources such as feces, dander, urine, feed, skin, and hair, which disperse into the air and generate significant microbial aerosols [2]. In this highly polluted environment, particulate matter (PM) is not the only concern, as heavy metals, toxic gases, and microorganisms can also pose health risks by entering the body. [3]. Among these pollutants, PM and ammonia (NH_3_) are the most prominent. Studies have indicated that chronic exposure to PM is a significant factor in the development of respiratory conditions, such as pneumonia and bronchitis. [4]. Furthermore, high concentrations of NH_3_ may cause serious damage to the eyes, nose, and skin [5]. These pollutants not only pose a threat to workers and animals in the facility, but can also be released into the external environment through ventilation systems, contributing to the deterioration of regional air quality.

To effectively mitigate the pollutants in animal facilities, improving the ventilation efficiency of animal facilities is a crucial step. This requires installing advanced ventilation systems and utilizing suitable filter materials that can significantly lower pollutant levels, thus mitigating negative health impacts on both animals and workers [6]. Currently, activated carbon and glass fibers are the primary filter materials used for air purification in animal-rearing environments [7]. Activated carbon is widely employed for its large surface area and strong adsorption capacity, making it highly effective in capturing NH_3_ and volatile organic compounds (VOCs) [8]. Glass fibers, on the other hand, are valued for their structural durability and are primarily used to physically block bacteria and trap larger airborne particles. However, it is difficult for these materials to simultaneously capture PM and adsorb NH_3_ [9,10,11]. Additionally, due to the high concentration of pollutants in animal facilities, it is also essential to have sufficient dust-holding capacity (DHC) in addition to fulfilling basic filtering requirements. Filters with high DHC can more effectively extend the service life of the filtration system, reducing the frequency of maintenance and replacement, and reducing operating costs [12]. Therefore, developing an integrated filter material capable of simultaneously achieving all these functions represents a significant challenge for both the research community and industry practitioners.

Aerogels are one of the most attractive materials, with ultra-low density, minimal pore size, and high porosity, making them highly suitable for air purification [13,14]. The filtration performance of aerogels is significantly influenced by their morphological structure. For example, Qiao et al. (2020) enhanced air permeability and maintained over 99.8% filtration efficiency by incorporating polyvinylidene fluoride (PVDF) into a 3D crosslinked polyimide (PI) network [15]. Su et al. (2024) fabricated an oxidized “grooved and secondary pore”-structured polyaryl thioether sulfone (O-GPPASS) aerogel, achieving PM_0.3_ filtration efficiency for 99.7% and a low pressure drop of 17.2 Pa [16]. Moreover, compared with two-dimensional filtration media, aerogels’ three-dimensional structures enable them to capture more particles, substantially increasing dust-holding capacity and extending the lifespan of filtration materials [12,17,18]. Aerogels integrated with metal–organic frameworks (MOFs) further enhance gas adsorption capabilities, demonstrating effectiveness in capturing formaldehyde, methane, carbon dioxide, and hydrogen sulfide [10,19,20,21,22,23]. Notably, diverse functionalized variants of UiO-66, especially UiO-66-NH_2_, exhibit exceptional NH_3_ removal performance in environments [24,25,26]. However, despite these advancements demonstrating the potential of aerogels, existing studies often focus on optimizing either PM filtration efficiency or gas adsorption. Therefore, developing aerogels that perform exceptionally in both aspects remains a significant challenge.

In nature, needles of the *Pinus sylvestris var. mongolica Litv.* (PS) exhibit unique structural advantages for dust retention. Their groove structures increase the specific surface area, enhancing their effectiveness in intercepting PM. Additionally, the irregular arrangement of the needles effectively reduces air resistance. When dust-laden airflow passes through PS, these varying orientations generate vortices that efficiently capture particulates while maintaining low airflow resistance. Based on this phenomenon, and combined with the exceptional adsorption capabilities of UiO-66-NH_2_ nanoparticles, we developed a high-performance biomimetic aerogel in this study. The research focuses on two primary objectives: (1) optimizing fiber structures by adjusting the polylactic acid (PLA)-to-gelatin ratio during the electrospinning process to prepare biomimetic fibers, and (2) incorporating UiO-66-NH_2_ nanoparticles into these fibers, utilizing freeze-drying technology to assemble one-dimensional fiber into three-dimensional aerogels with low airflow resistance and efficient NH_3_ adsorption. The filtration efficiency, dust-holding capacity, NH_3_ adsorption performance, and elasticity of the developed aerogel were systematically evaluated. These assessments aimed to establish whether this innovative material represents a viable solution for air purification in heavily polluted environments. Furthermore, this work aimed to contribute to advancing biomimetic filtration technologies, supporting both industrial pollution control efforts and environmental sustainability initiatives.

## 2. Materials and Methods

### 2.1. Materials

Zirconium chloride (ZrCl_4_, 98%), 2-amino-1,4-dicarboxylic acid (BDC-NH_2_, 98%), poly (lactic acid) (PLA, *M_W_* = 110,000), 1,1,1,3,3,3-hexafluoro-2-propanol (HFIP, 99.5%), and *N*,*N*-dimethylformamide (DMF, AR, ≥99.5%) were obtained from Shanghai Macklin Biochemical Co., Ltd. (Shanghai, China). Gelatin (Type A) was provided by Shanghai yuanye Bio-Technology Co., Ltd. (Shanghai, China). Tert-Butyl alcohol was obtained from Tianjin Xinbote Chemical Co., Ltd. (Tianjin, China). Methanol was bought from Tianjin Fuyu Fine Chemical Co., Ltd. (Tianjin, China).

### 2.2. Fabrication of Biomimetic Fibers

The preparation of the biomimetic fibers was undertaken as follows: first, PLA was dissolved in HFIP to prepare 5 wt% PLA solution and stirred at 60 °C for 3 h. At the same time, gelatin was dissolved in HFIP to prepare 8 wt% gelatin solution, and this mixture was also stirred at 60 °C for 3 h. Then, the PLA and gelatin solutions were mixed at PLA and gelatin mass ratios of 6:0, 5:1, 4:2, 3:3, 2:4, 1:5, and 0:6, and stirred to obtain solutions in a homogeneous state. These various solutions were fed to the electrospinning machine (Elite, Beijing Ucalery Technology and Development Co., Ltd., Beijing, China) with a positive voltage of 15 KV, voltage pressure of 5 KV, flow rate of 1.5 mL/h, and distance between the spinneret and the collector of 15 cm, to obtain membranes labelled as P6, P5G1, P4G2, P3G3, P2G4, P1G5, G6, respectively.

### 2.3. Fabrication of UiO-66-NH_2_@PLA/Gelatin Nanofibrous Aerogels

The synthesis of UiO-66-NH_2_ nanoparticles was carried out in accordance with the method detailed in a previous publication [27,28]. Figure 1a shows the preparation process of the aerogels. Firstly, the P1G5 nanofiber membrane was cut into small pieces of about 1 cm × 1 cm, and 0.4 g of P1G5 nanofibers were put into 20 mL of tert-butyl alcohol. Then the mixture was uniformly ultrasonicated at 600 W for 2 min with an ultrasonic cell crusher (SCIENTZ-IID, Ningbo Xinzhi Biotechnology Co., Ltd., Ningbo, China). Meanwhile, different amounts of UiO-66-NH_2_ at 0, 0.02, 0.04, 0.06, and 0.08 g were homogeneously dispersed into 20 mL of tert-butyl alcohol, then added into the P1G5 nanofibers/tert-butyl alcohol mixture and homogeneously dispersed again by an ultrasonic cell crusher. Next, 10 mL of the above homogeneous nanofiber mixtures were poured into 38.5 mm diameter molds and cooled to −30 °C for 12 h. The samples were subsequently freeze-dried in a freeze-dryer (SCIENTZ-12N, Ningbo Xinzhi Biotechnology Co., Ltd., Ningbo, China) for 24 h. Finally, the obtained 3D aerogels were cross-linked in a muffle furnace (MF-1700C-II, Anhui BeiYiKe Equipment Technology Co., Ltd., Hefei, China) at 140 °C for 2 h. The obtained aerogel samples were labelled as PG, UPG-5, UPG-10, UPG-15, and UPG-20, according to UiO-66-NH_2_ content relative to P1G5 nanofibers (0%, 5%, 10%, 15%, and 20%). Additionally, pure PLA aerogel and Gelatin aerogel were prepared using P6 nanofiber and G6 nanofiber and used as reference. Figure 1b shows the schematic diagram of air purification with UPG aerogels.

### 2.4. Characterization

A scanning electron microscope (SEM) (Su-70, Hitachi, Tokyo, Japan) was used to characterize the morphologies of the prepared membranes and aerogels. The X-ray diffraction patterns of the samples were analyzed using XRD (6100, Shimadzu, Kyoto, Japan), while sample chemical compositions was determined using FT-IR (iS50, Thermo Nicolet Corporation, Madison, WI, USA). The surface chemical states and elemental compositions were examined using XPS (Escalab250Xi, ThermoFisher, Waltham, MA, USA), and TGA/DTG (HTG-1, Henven, Beijing, China) was used to study the thermal stability of the aerogels under a N_2_ atmosphere. N_2_ adsorption desorption curves were measured using an advanced gas adsorption and micropore analyzer (BSD-660 M, BSD Instrument Inc., Beijing, China). The specific surface area and pore size distribution were analyzed using the BET methods and NLDFT methods. An automated gas sorption analyzer (Autosorb iQ, Quantachrome Instrument Inc., Boynton Beach, FL, USA) was used to analyze the NH_3_ adsorption capacity.

### 2.5. Filtration Performance Measurement

The filtration performance was evaluated utilizing the test platform (Appendix A) with a controlled flow rate of 5.3 cm/s. The filtration efficiency of PM (µg/cm^3^) was calculated using Equation (1) [29]:(1)$$\eta \left(PM\right)=\frac{C_{up}\left(PM\right)-C_{down}(PM)}{C_{up}\left(PM\right)}$$
where $C_{up}\left(PM\right)$ and $C_{down}(PM)$ represent the upstream and downstream concentrations of PM_1_ (particle sizes less than 1 µm), PM_2.5_ (particle sizes less than 1 µm), and PM_10_ (particle sizes less than 1 µm), respectively.

To comprehensively assess the filtration performance, the quality factor (QF, Pa^−1^) was calculated using Equation (2) [30]:(2)$$QF=\frac{-In(1-\eta )}{\Delta P}$$
where Δ*P* (Pa) is the pressure drop of the tested membranes/aerogels.

The DHCs of the membranes/aerogels were measured using the same filter holder (Appendix A) under a flow rate of 10.6 cm/s. The measurement was terminated once the pressure drop of membranes/aerogels reached the preset industrial threshold of 1000 Pa. The DHCs of membranes/aerogels was calculated using Equation (3) [31]:(3)$$D=(M_{t}-M_{0})/A$$
where *M_t_* (g) and *M*_0_ (g) are the mass of the aerogels with accumulated PM and the clean aerogels, respectively, and *A* (m^2^) is the effective area of the aerogels. All tests were conducted at 25 ± 5 °C.

## 3. Results and Discussion

### 3.1. Characterization of Bionic Ribbon PLA/Gel Fiber Membrane

The physical morphology of nanofibers had an important influence on their filtration performance [32]. Conventional nanofibers, characterized by their smooth and bead-free structure, seem to be inadequate in terms of filtration performance owing to their conventional interception mechanisms [33]. In nature, PS has been identified as an effective approach to mitigate PM in the atmosphere, primarily due to the presence of numerous “grooved” stripes in the microstructure of its needles [34]. Figure 2a shows an optical photograph of PS, with the inset showing a 3D ultra-depth-of-field digital image of PS needles, which clearly reveals the striped grooves and pore bands on the needle surface. Additionally, the SEM image in Figure 2b highlights the prominent “grooved” structures on the PS needles. In this study, the structure and morphology of bionic grooved ribbon fibers were controlled by varying the PLA-to-gelatin mass ratio in the precursor solutions using electrospinning technology. As shown in Figure 2c, pure PLA nanofibers display a cylindrical shape with some small protrusions on the surface. With an increasing concentration of gelatin (from P5G1 to P3G3), the rod-like morphology of the nanofibers remains largely unaffected, while the surface protrusions become more pronounced, aligning in the direction of the fibers, as illustrated in Figure 2d–f. When the gelatin content exceeds that of PLA, the resulting fibers exhibit a cylindrical shape with long grooves, as shown in Figure 2g. Notably, P1G5 morphology shows ribbon-like fibers with distinct long grooves, forming a bionic nano-grooved ribbon fiber ranging from approximately 166 nm to 261 nm, as depicted in Figure 2h. Conversely, pure gelatin fibers exhibit a smooth ribbon-like morphology, as shown in Figure 2i. These results demonstrate that in the mixed solution system composed of PLA and gelatin, reducing the PLA content promotes the formation of vertical grooves on the fiber surface, while increasing gelatin content gradually transforms the morphology to ribbon-like fibers.

The formation mechanism of the bionic grooved ribbon fibers is further analyzed, as depicted in Figure 2j. In the mixed solution system, the heterogeneous evaporation of solvent plays a key role in forming the nano-grooved ribbon fiber morphology [33,35,36]. Firstly, the solution conductivity directly impacts the morphology of electrospun nanofibers [37]. The increased conductivity alters the electric field intensity and accelerates solvent evaporation, thereby affecting the formation of uniquely shaped fibers [38]. Compared to the pure PLA precursor, the addition of gelatin markedly improves the conductivity of the precursor solutions (Table 1), which promotes the formation of ribbon-like fibers during the electrospinning. Additionally, the concentration of gelatin serves as a key factor in controlling fiber morphology. As the gelatin concentration increases, the fibers develop longer grooves, gradually transitioning from rod-like to ribbon-like structures. The amino and carboxyl groups present in gelatin form hydrogen bonds with the hydroxyl or carboxyl groups in PLA, resulting in a cross-linked system. Higher gelatin content increases the degree of cross-linking, leading to enhanced polymer chain entanglement. Under these conditions, the solvent diffuses gradually from the interior of the jet to the surface, resulting in the formation of nano-grooved ribbon fibers.

### 3.2. Characterization of PLA/Gel/MOF Nanofibrous Aerogels

The bionic nano-grooved ribbon fibers were subsequently used to prepare PG aerogels and UPG aerogels, with the latter doped with varying masses of UiO-66-NH_2_. The SEM analysis of the synthesized UiO-66-NH_2_ nanoparticles (Appendix A) confirmed that they have a distinctive octahedral morphology, consistent with established crystal structures reported in the literature [39,40]. The microstructures of aerogels with different UiO-66-NH_2_ concentrations are presented in Figure 3a–e, revealing that all aerogels retain the same ribbon fiber morphology as the original PG nanofibers. With an increase in the concentration of UiO-66-NH_2_ nanoparticles, a greater number of nanoparticles are observed on the ribbon fibers. Furthermore, the bionic nano-grooved ribbon fibers exhibit a randomly oriented arrangement within the three-dimensional aerogel, allowing both the main surfaces and the edges of the ribbon-like fibers to be visible. This multidirectional alignment contributes to reducing airflow resistance. As presented in Figure 3f (magnified view of UPG-10), the UiO-66-NH_2_ was uniformly dispersed onto the surface of the ribbon fibers. EDS elemental mapping analysis of UPG-10 aerogel confirms the presence and uniform distribution of these elements (Figure 3g). Additionally, a piece of UPG-10 aerogel with a density of about 14.9 ± 0.2 mg/cm^3^ UPG-10 aerogel can be effortlessly placed onto the stamen of a flower, as shown in Figure 3h, highlighting its lightweight nature.

To investigate the effect of doping concentration on PG aerogels, a series of UPG aerogels with different UiO-66-NH_2_ contents were characterized. Figure 4a displays the XRD spectrum of the aerogels before and after mixing with UiO-66-NH_2_. The peak at 20.6° for the PG sample can be attributed to the characteristic peak of gelatin, but the characteristic peak of PLA could not be seen, probably due to the low PLA content [41]. Typical strong diffraction peaks correspond to the (111) and (222) lattice planes of UiO-66-NH_2_, respectively. The characteristic peaks of UiO-66-NH_2_ were also observed in the diffraction patterns of the composite aerogels. Figure 4b presents the FTIR spectra of the UiO-66-NH_2_, PG, and UPG aerogels. The characteristic absorption bands at approximately 620 cm^−1^ and 770 cm^−1^ are attributed to Zr-O bonds in UiO-66-NH_2_ [42,43]. The peak at 3100–3600 cm^−1^ corresponds to the stretching vibration of O–H groups of PLA in the PG aerogel and UiO-66-NH_2_, as well as the N–H stretching in the gelatin in the PG aerogel [44]. The peaks at 1644 cm^−1^ and 1536 cm^−1^ represent the amide I and amide II bands of the PG aerogel, respectively [44,45]. The peak at 1750 cm^−1^ is indicative of the –CHO stretching vibration in PLA, while the range from 2850 to 2950 cm^−1^ denotes the aliphatic C–H stretching found in both gelatin and PLA [46]. As shown in the survey spectrum of Figure 4c, UPG-x exhibits all elemental peaks of both PG and UiO-66-NH_2_, including C 1s, N 1s, O 1s, and Zr 3d. An increase in UiO-66-NH_2_ concentration enhances the intensities of C 1s and Zr 3d peaks while reducing those of O 1s and N 1s, reflecting a higher proportion of C and Zr elements and a lower proportion of O and N elements, which is consistent with the compositional data in Appendix A. The high-resolution XPS spectrum for Zr 3d in UiO-66-NH_2_/UPG-x features two distinct peaks at 185.23 eV and 182.84 eV for the 3d_3/2_ and 3d_5/2_ levels, respectively, consistent with those in pure UiO-66-NH_2_, as depicted in Figure 4d. The characteristic peak intensity of UiO-66-NH_2_ progressively increased as its loading ratio increased. Additionally, Appendix A presents the high-resolution XPS spectra of C 1s, N 1s, O 1s, and Zr 3d over UPG-10 respectively [47,48,49,50]. The presence of these characteristic peaks in UPG aerogels verifies the successful incorporation of UiO-66-NH_2_ into the PG aerogels.

Figure 4e,f illustrates the TGA curves and DTG curves of UiO-66-NH_2_, PG aerogel, and UPG aerogels. The weight loss of the PG aerogel at nearly 100 °C was primarily caused by the evaporation of water from gelatin. The subsequent thermal decomposition stages involve the degradation of PLA and gelatin, as well as the gasification of their decomposition products [51]. For UiO-66-NH_2_, three stages of weight loss were observed. Below 200 °C, the weight loss was associated with solvent evaporation. Between 200 and 300 °C, the second stage occurred due to the dihydroxylation of Zr_6_O_4_(OH)_4_ clusters transforming into Zr_6_O_6_. Beyond 400 °C, the third stage corresponded to the decomposition of organic groups [42]. Notably, with the increase in the mass fraction of UiO-66-NH_2_, the residual weight percentage of the aerogels exhibited a gradual upward trend. Calculations based on Appendix A show that the actual loading rates of UiO-66-NH_2_ in UPG-5, UPG-10, UPG-15, and UPG-20 were 5.8 wt%, 9.8 wt%, 13.8 wt%, and 21.8 wt%, respectively, as detailed in Appendix A. The percentage of UiO-66-NH_2_ in the suspensions is lower than in the corresponding UPG aerogels, mainly due to the loss of PG fibers caused by the unstable structure of the aerogels during the freeze-drying process.

To examine the impact of UiO-66-NH_2_ content on the pore structure of aerogels, N_2_ adsorption–desorption experiments were conducted on UiO-66-NH_2_ and a series of aerogels, as shown in Figure 4g–i and Table 2. The results revealed that UiO-66-NH_2_ exhibits a high volume of micropores (Appendix A). In particular, as the UiO-66-NH_2_ content increases, both the specific surface area and the micropore volume of the aerogels increase. The improvement in microporosity demonstrates that a higher UiO-66-NH2 content effectively enhances the pore structure of the aerogels, making them more suitable for use in gas adsorption and separation.

### 3.3. Air Purification Performance of the Aerogels

The air purification performance of the aerogels was comprehensively evaluated, focusing on PM, NH_3_ adsorption, and DHC. In accordance with the air filtration standards (EN779:2012) [52], this study evaluated the filtration performance of the aerogels. The tests were conducted at a flow rate of 5.3 cm/s, using dust collected from animal farms as the target airborne particulate matter (PM). Figure 5a shows the filtration efficiencies and pressure drops for PG aerogel and UiO-66-NH_2_-doped PG aerogels (UPG) at different concentrations. The results showed a similar trend in filtration efficiency across various particle sizes. Taking PM_1_ as an example, the filtration efficiency and pressure drop gradually increased from PG to UPG-10, with UPG-10 achieving the highest filtration efficiency of 99.24% and a pressure drop of 95 Pa. In contrast, UPG-15 and UPG-20 exhibited reductions in both filtration efficiency and pressure drop. This decline may stem from the structural instability of the aerogel fibers caused by the addition of larger quantities of UiO-66-NH_2_ nanoparticles. Notably, the UPG-10 exhibited the highest QF of 0.051 (for PM_1_) compared with other aerogels, as shown in Figure 5b. Moreover, compared with cylindrical fiber aerogel (PLA) and ribbon fiber aerogel (gelatin), the UPG-10 aerogel yielded the optimal filtration performance, as depicted in Figure 5c. This advantage is mainly attributed to the uniquely designed bionic “groove” ribbon fiber structure and the rougher surface caused by the UiO-66-NH_2_ of the UPG-10 aerogel, making it more effective in capturing PM. In addition, the UPG-10 aerogel demonstrated a significant advantage in filtration performance compared to other reported filtration materials (Figure 5d). To further assess the stability of the UPG-10 aerogel in high-concentration particulate environments, a cyclic test was performed to remove dust with an initial concentration of approximately 1000 µg/m^3^. Over 20 repeated filtration cycles, the time required to reduce the particulate concentration to 50 µg/m^3^ remained nearly constant, demonstrating the exceptional stability and efficiency of UPG-10 aerogels in filtering high-concentration particulate matter, as illustrated in Figure 5e.

Figure 5f illustrates the NH_3_ adsorption and desorption isotherms (at 25 °C) for both the PG aerogel and the UPG-10 aerogel. The maximum adsorption capacities of the PG aerogel and UPG-10 aerogel reached approximately 99.89 cm^3^/g and 68.43 cm^3^/g, respectively. This significant disparity in adsorption capacity is largely due to the integration of UiO-66-NH_2_ nanoparticles into the UPG-10 aerogel. The addition of UiO-66-NH_2_ substantially increased the specific surface area and pore volume of the aerogel, creating more active sites for NH_3_ adsorption. Furthermore, the -NH_2_ functional groups present in UiO-66-NH_2_ were crucial in improving interactions with NH_3_ by forming strong hydrogen bonds, as depicted in Figure 5g. This combination of an enlarged surface area and robust chemical interactions underscores the effectiveness of UiO-66-NH_2_ in improving the NH_3_ adsorption performance of the aerogels [53].

**Figure 5 polymers-17-01234-f005:**
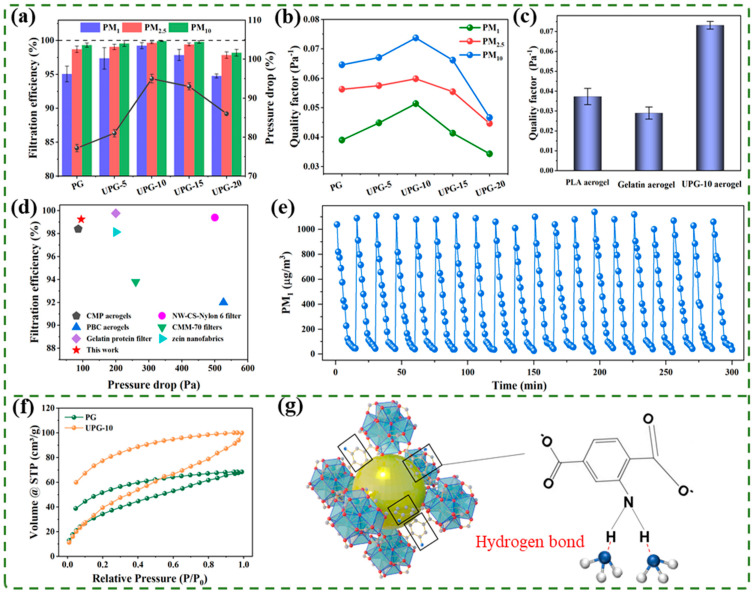
Air purification performance of the aerogels. (**a**) Filtration efficiency and pressure drop of the aerogels. (**b**) Quality factor of the aerogels. (**c**) Quality factor comparison among PLA aerogel, gelatin aerogel, and UPG-10 aerogel. (**d**) Comparison of the filtration performance of the UPG-10 aerogel with other reported filtration materials [11,54,55,56,57,58]. (**e**) Cyclic filtration test for PM_1_ concentration variation over time. (**f**) The NH_3_ adsorption isotherm of the PG aerogel and UPG-10 aerogel at 25 °C. (**g**) Schematic illustrating the adsorption of NH_3_ by UiO-66-NH_2_ via hydrogen bonding with its NH_2_ groups.

To better understand the advantages of three-dimensional aerogel structures in filtration applications, the structures of PG membranes and UPG-10 aerogels were analyzed. Figure 6a illustrates the overall layered distribution of the PG membrane, characterized by tightly packed fibers forming a dense structure. While this dense arrangement enhances particle capture, it may also result in higher pressure drop (Figure 6c). In contrast, Figure 6b reveals the three-dimensional porous fiber network structure of the UPG-10 aerogel, where the pores between fibers were significantly larger than those in the membrane. Due to the random distribution of fibers, particles repeatedly collide with fiber surfaces during their motion, and mutual collisions further alter their trajectories (Figure 6d). The filtration performance of the PG membrane and UPG-10 aerogel is shown in Figure 6e. While their filtration efficiencies for PM are similar, the UPG-10 aerogel demonstrates a 40 Pa lower pressure drop compared to the PG membrane, resulting in a higher quality factor (QF) for the aerogel, as shown in Figure 6f. This improved performance is primarily owing to the higher porosity and lower density of the UPG-10 aerogel (Figure 6g). The high porosity effectively reduces pressure drop, while the increased thickness ensures that the UPG-10 aerogel maintains high filtration efficiency, achieving superior overall filtration performance. Additionally, this structure greatly enhances the DHC of the UPG-10 aerogel. Figure 6h shows the filtration efficiency of PG aerogels and UPG-10 aerogels over a usage cycle, indicating that their filtration efficiencies did not differ significantly at different stages and gradually increased to nearly 100%. However, a significant difference was observed in their DHC. As shown in Figure 6i, UPG-10 aerogels exhibited an exceptional DHC of 147 g/m^2^, which is three times higher than that of PG membranes. Furthermore, Figure 6j provides a comparison of the DHC of UPG-10 aerogels with other reported studies, highlighting their superior performance [10,17,28,31]. The UPG-10 aerogel exhibits a distinct advantage due to its excellent pore structure and high porosity, enabling a longer service life.

### 3.4. Mechanical Performance of UPG-3 Aerogel

Good mechanical performance is crucial for an effective air filter. As shown in Figure 7a,b, UPG-10 aerogel exhibits high flexibility in both compression and bending tests, withstanding significant deformation and fully recovering its original shape. The compressive stress–strain curves at 20%, 40%, and 60% strain levels (Figure 7c) further highlight its excellent resilience under varying strains. The maximum stress exhibited a significant increase with strain, attaining values of 0.54 kPa at 20% strain and 1.75 kPa at 60% strain. Figure 7d presents the maximum compressive stress and plastic deformation across multiple compression cycles (1st, 10th, 20th, and 30th). With increasing cycles, the maximum compressive stress shows a slight decrease, while plastic deformation gradually increases. Even after 30 cycles, the aerogel maintains a high compressive stress level of 1.47 and recovers over 80% of its deformation, demonstrating strong fatigue resistance and elastic stability, as shown in Figure 7e. These properties highlight the potential of UPG-10 aerogel for air filtration applications. Figure 7f illustrates the elastic mechanism of the UPG-10 aerogel. When the aerogel is exposed to external compressive stress, individual nanofibers experience normal stress (F_1_) perpendicular to the applied force. At the same time, adjacent fibers exhibit a tendency to move in opposing directions, leading to the generation of shear forces (F_2_) at their contact points. The superior elasticity of the aerogel is primarily attributed to these stable bonding points formed through high-temperature crosslinking. These bonding points play a crucial role by supporting the interconnections between nanofibers, enabling the network to maintain and recover its three-dimensional structure under applied stress. In contrast, the uncross-linked aerogels failed to recover after compression (Appendix A) due to the absence of cross-links, which rendered the nanofiber network incapable of sustaining the 3D structure under external forces.

## 4. Conclusions

In this study, a high-performance biomimetic aerogel was developed, inspired by the dust-retention properties of *Pinus sylvestris var. mongolica* needles. By combining electrospinning and freeze-forming technologies, we successfully fabricated a three-dimensional grooved ribbon-fiber aerogel incorporating UiO-66-NH_2_ nanoparticles (UPG). This innovative design integrates structural advantages with functional enhancements, demonstrating outstanding filtration performance and mechanical stability. Among the various formulations evaluated, the UPG-10 aerogel achieved a filtration efficiency of 99.24% for PM_1_ with a low pressure drop of 95 Pa, outperforming conventional PLA and gelatin aerogels. Its dust-holding capacity was three times higher than that of PG fiber membranes, ensuring an extended service life in high-concentration particle environments. The incorporation of UiO-66-NH_2_ nanoparticles significantly enhanced NH_3_ adsorption capacity through increased surface area and robust hydrogen bonding interactions, resulting in a 45.97% improvement compared to conventional PG aerogels. Additionally, mechanical evaluations demonstrated the excellent elasticity and resilience of the UPG-10 aerogel, retaining over 80% of its original shape after 30 compression cycles. Overall, the UPG-10 aerogel demonstrates a synergistic combination of high filtration efficiency, large dust-holding capacity, effective NH_3_ adsorption, and mechanical robustness. These properties make it a promising material for air purification, and offer insights for advancing biomimetic filtration technologies.

## Figures and Tables

**Figure 1 polymers-17-01234-f001:**
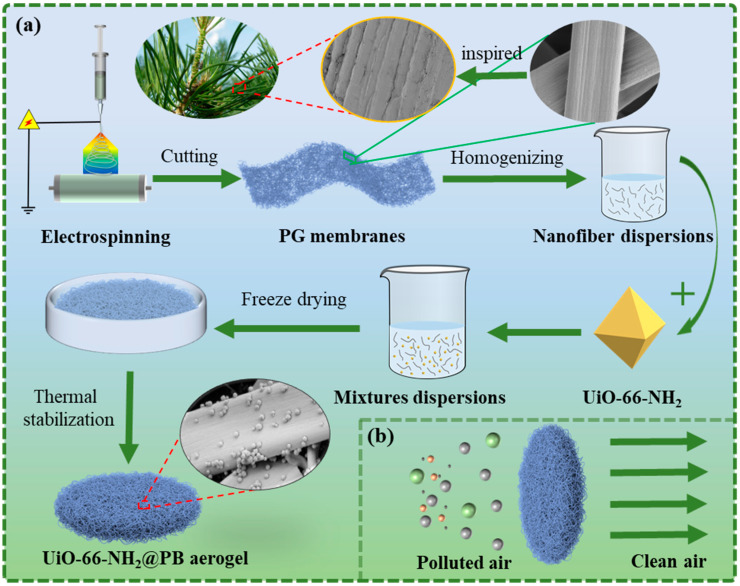
Schematic illustration of the preparation process (**a**) and use for filtration (**b**) of bionic UPG aerogels.

**Figure 2 polymers-17-01234-f002:**
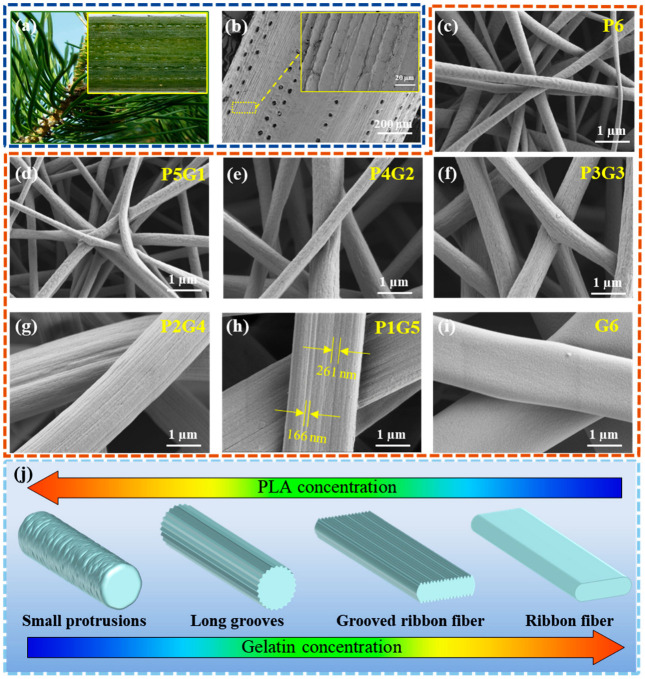
Characterization and formation mechanism of the bionic striped ribbon fibers. (**a**) Optical photograph and 3D ultra-depth-of-field digital photograph of PS; (**b**) SEM images of PS needles; SEM images of the (**c**) P6, (**d**) P5G1, (**e**) P4G2, (**f**) P3G3, (**g**) P2G4, (**h**) P1G5, and (**i**) G6 electrospun fibers; and (**j**) formation mechanism of the bionic striped ribbon fibers.

**Figure 3 polymers-17-01234-f003:**
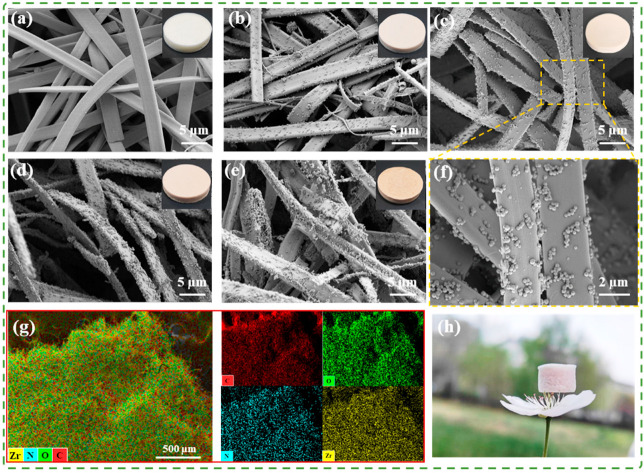
Morphological characterization of the aerogels. SEM of (**a**) PG, (**b**) UPG-5, (**c**) UPG-10, (**d**) UPG-15, (**e**) UPG-20, and the insets are their optical photographs. (**f**) Magnified view of UPG-10. (**g**) EDS elemental mapping image of the UPG-10 aerogel. (**h**) Optical photo of UPG-10 on the stamen of a flower.

**Figure 4 polymers-17-01234-f004:**
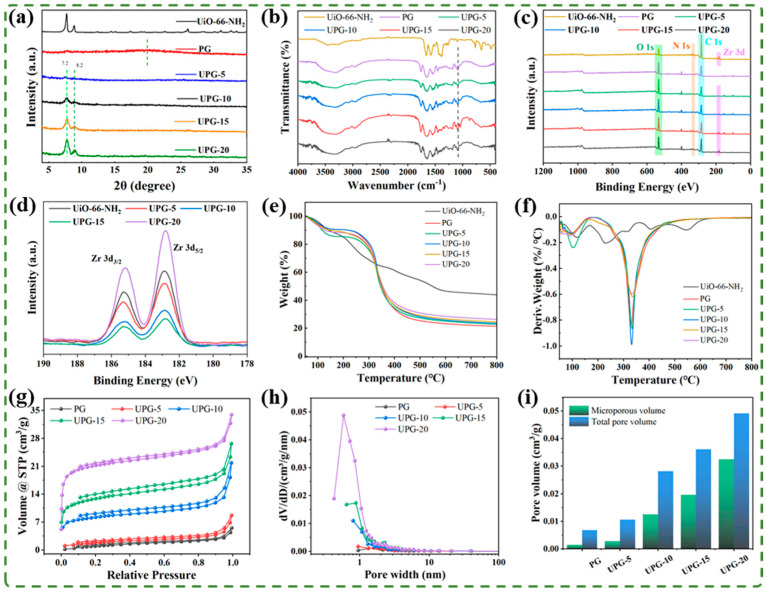
Characterization of the aerogels. (**a**) XRD, (**b**) FTIR, (**c**) XPS, (**d**) XPS spectra of Zr, (**e**) TGA, (**f**) DTG of the aerogels, (**g**) N_2_ adsorption–desorption curves, (**h**) pore size distribution, and (**i**) pore volume of the different aerogels.

**Figure 6 polymers-17-01234-f006:**
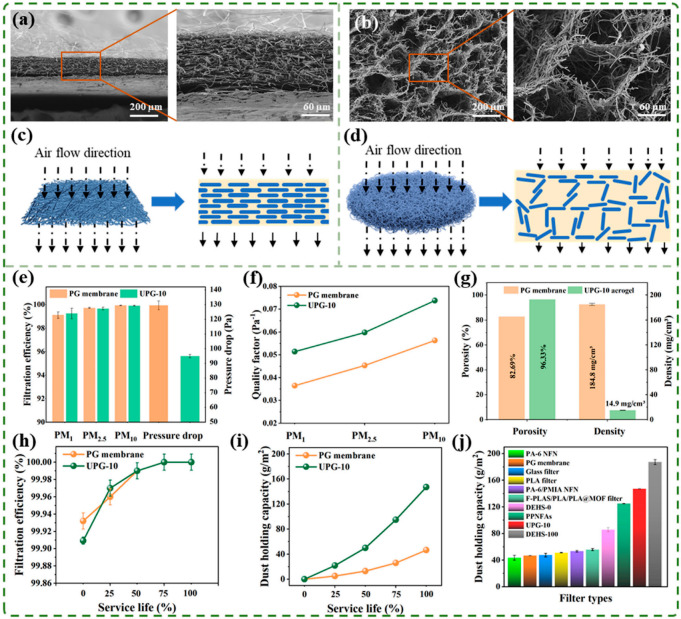
Comparison of the filtration performance between PG membrane and UPG-10 aerogel. SEM images and schematic diagram of the cross-section of (**a**,**c**) PG membrane and (**b**,**d**) UPG-10 aerogel. (**e**) Filtration efficiency, pressure drop, and (**f**) quality factor of PG membrane and UPG-10 aerogel for different particulate sizes. (**g**) Comparison of porosity and density of PG membrane and UPG-10 aerogel. (**h**) Filtration efficiency and (**i**) dust-holding capacity for PG Membrane and UPG-10 aerogel during a usage cycle. (**j**) Comparison of DHC between filtration materials prepared in this study with other reported studies [10,17,28,31].

**Figure 7 polymers-17-01234-f007:**
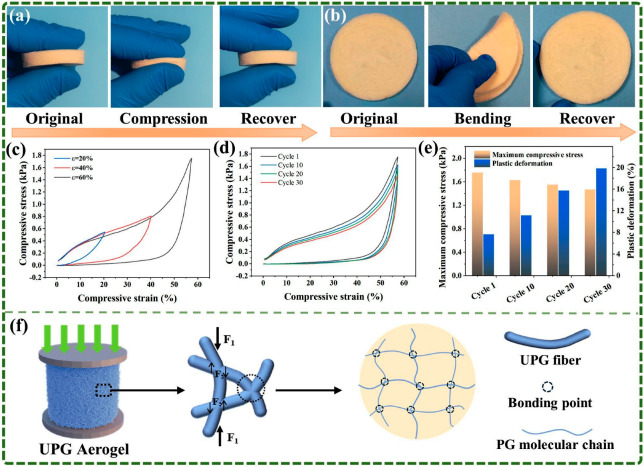
Mechanical properties and elastic mechanism of UPG-10 aerogel. High flexibility of UPG-10 aerogel demonstrated by (**a**) compressing test and (**b**) bending test. (**c**) Compressive stress–strain curves of the UPG-10 aerogel at different strains. (**d**) Compressive stress–strain curves under different compression cycles. (**e**) Maximum compressive stress and plastic deformation across compression cycles. (**f**) Schematic diagram of the internal stress conditions of UPG aerogel under the action of external force.

**Table 1 polymers-17-01234-t001:** Conductivity of electrospinning precursors.

Electrospinning Precursors	P6	P5G1	P4G2	P3G3	P4G2	P5G1	G6
Conductivity (µS cm^−1^)	0.031 ± 0.002	0.517 ± 0.002	0.707 ± 0.002	0.905 ± 0.001	1.168 ± 0.001	1.321 ± 0.002	1.528 ± 0.003

**Table 2 polymers-17-01234-t002:** Comparison of pore characteristics of the aerogels and UiO-66-NH_2_ nanoparticles.

Samples	Average Pore Diameter (nm)	BET Surface Area (m^2^/g)
PG	5.48	5.55
UPG-5	7.09	7.10
UPG-10	4.32	30.08
UPG-15	3.37	47.30
UPG-20	2.52	81.26
UiO-66-NH_2_	2.49	814.90

## Data Availability

The original contributions presented in this study are included in the article. Further inquiries can be directed to the corresponding authors.

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
