# Peer review of "Biomimetic Grooved Ribbon Aerogel Inspired by the Structure of Pinus sylvestris var. mongolica Needles for Efficient Air Purification"

_polymers, 2025, doi:10.3390/polym17091234_

Round 1
Reviewer 1 Report
Comments and Suggestions for Authors
Summary: The authors have presented PLA and Gelatin based membranes for air purification. The microstructure of the membrane fibers is inspired by the Pinus sylvestris var. mongolica needles.
Overall Comments: The author have showcased high-quality work in this manuscript. I found the work here described and presented in detail. The experiments and their design were very extensive and simulated product testing similar to real world applications. Additionally, I found the comparison to the materials from other studies very thoughtful and helped make the work comprehensive.
Minor Comments/Suggestions: I do still have minor comment for the authors, where this manuscript needs improvement.
- L-128: correct to ‘shows’ instead of shown
- L-130 Tert-butyl is not a compound, please replace it with appropriate compound
- The fabrication of U-PG (section 2.3) is a bit confusing can you please describe it in more detail.
- The rationale behind choosing P1G5 for fabricating U-PG is not explained in the manuscript nor supported by properties or performance data. Can the authors please explain their reasoning for choosing that particular composition over other. Maybe support their reasoning with some data.
- I also want to point out that the needles of PS have a circular grooved geometry. In contrast the authors chose a flat grooved ribbon for this work. I have not being able to rationalize their reasoning for choosing a flatter grooved structure over circular grooved structure.
- How did the authors determine the actual level of doping with nano-particles for the U-PG membranes. Clearly there must have been some loss of nano-particles. I find it difficult to believe that all the intentioned nano-particles were attached to the fiber surface.
- The instrument info and method for mechanical characterization is missing.
Author Response
Comments 1:
Summary: The authors have presented PLA and Gelatin based membranes for air purification. The microstructure of the membrane fibers is inspired by the Pinus sylvestris var. mongolica needles.
Overall Comments: The author have showcased high-quality work in this manuscript. I found the work here described and presented in detail. The experiments and their design were very extensive and simulated product testing similar to real world applications. Additionally, I found the comparison to the materials from other studies very thoughtful and helped make the work comprehensive.
Minor Comments/Suggestions: I do still have minor comment for the authors, where this manuscript needs improvement.
L-128: correct to ‘shows’ instead of shown
L-130 Tert-butyl is not a compound, please replace it with appropriate compound
The fabrication of U-PG (section 2.3) is a bit confusing can you please describe it in more detail.
The rationale behind choosing P1G5 for fabricating U-PG is not explained in the manuscript nor supported by properties or performance data. Can the authors please explain their reasoning for choosing that particular composition over other. Maybe support their reasoning with some data.
I also want to point out that the needles of PS have a circular grooved geometry. In contrast the authors chose a flat grooved ribbon for this work. I have not being able to rationalize their reasoning for choosing a flatter grooved structure over circular grooved structure.
How did the authors determine the actual level of doping with nano-particles for the U-PG membranes. Clearly there must have been some loss of nano-particles. I find it difficult to believe that all the intentioned nano-particles were attached to the fiber surface.
The instrument info and method for mechanical characterization is missing.
Response 1: We would like to express our sincere gratitude to the reviewer for their thorough evaluation and constructive feedback on our manuscript. We highly appreciate the positive comments regarding the quality of our work and the detailed presentation of our experiments and results. Below, we address the minor comment raised by the reviewer in detail.
Point 1: L-128: correct to ‘shows’ instead of shown
Response 1: We thank the reviewer for pointing out this grammatical issue. We have corrected the sentence at line 128 by replacing "shown" with "shows" to ensure grammatical accuracy. The revised sentence now reads as follows: "Figure 1a shows the preparation process of the aerogels" in line 130 of the revised manuscript.
Point 2: L-130 Tert-butyl is not a compound, please replace it with appropriate compound
Response 2: We sincerely thank the reviewer for pointing out this issue. Upon reviewing the text, we realize that the term "tert-butyl" was used ambiguously in this context. In fact, we intended to refer to tert-butyl alcohol, which is a specific compound. To clarify this and ensure accuracy, we have revised the sentence at line 130 as follows: "…were put into 20 mL of tert-butyl alcohol" in line 130 and 136 of the revised manuscript.
Point 3: The fabrication of U-PG (section 2.3) is a bit confusing can you please describe it in more detail.
Response 3: We appreciate the reviewer’s valuable comment regarding the fabrication process of U-PG. To clarify the preparation procedure, we have expanded the description of each step below, providing additional details and explanations for better understanding: " Figure 1a shows the preparation process of the aerogels. Firstly, the P1G5 nanofibers membrane were
cut into small pieces of about 1 cm × 1 cm, and 0.4 g of P1G5 nanofibers were put into 20 mL of tert-butyl alcohol. Then the mixture was uniformly ultrasonicated at 600 W for 2 min with an ultrasonic cell crusher (SCIENTZ-IID, Ningbo Xinzhi Biotech-nology Co., Ltd., China). Meanwhile, different amounts of UiO-66-NH2 at 0, 0.02, 0.04, 0.06, 0.08 g were homogeneously dispersed into 20 mL of tert-butyl alcohol, then added into the P1G5 nanofibers/tert-butyl alcohol mixture and homogeneously dispersed again by an ul-trasonic cell crusher. Next, 10 mL of the above homogeneous nanofiber mixtures were poured into 38.5 mm diameter molds and cooled to -30 °C for 12 h. The samples were subsequently freeze-dried in a freeze-dryer (SCIENTZ-12N, Ningbo Xinzhi Biotechnology Co., Ltd., China) for 24 h. Finally, the obtained 3D aerogels were cross-linked in a muffle furnace (MF-1700C-II, Anhui BeiYiKe Equipment Technology Co., Ltd., China) at 140 °C for 2 h. The obtained aerogel samples were labelled as PG, UPG-5, UPG-10, UPG-15 and UPG-20 according to UiO-66-NH2 content relative to P1G5 nanofibers (0%, 5%, 10%, 15%, and 20%). Additionally, pure PLA aerogel and Gelatin aerogel were prepared by P6 nano-fiber and G6 nanofiber and used as reference. Figure 1b shows the schematic diagram of air purification with UPG aerogels" in section 2.3 of the revised manuscript.
Point 4: The rationale behind choosing P1G5 for fabricating U-PG is not explained in the manuscript nor supported by properties or performance data. Can the authors please explain their reasoning for choosing that particular composition over other. Maybe support their reasoning with some data.
Response 4: We thank the reviewer for their valuable comment regarding the rationale for choosing P1G5. Initially, P1G5 was selected based on qualitative observations of its favorable fiber morphology, which featured a grooved ribbon structure akin to a needle of Pinus sylvestris var. mongolica (PS) configuration. This choice was later supported by comparative filtration performance tests, where UPG aerogels made from P1G5 outperformed conventional gelatin and PLA aerogels, demonstrating superior filtration performance (Figure 5c). While we acknowledge the need for more comprehensive studies on other compositions, this will require extensive experimentation, which we plan to address in future work. We appreciate the reviewer’s feedback.
Point 5: I also want to point out that the needles of PS have a circular grooved geometry. In contrast the authors chose a flat grooved ribbon for this work. I have not being able to rationalize their reasoning for choosing a flatter grooved structure over circular grooved structure.
Response 5: We sincerely thank the reviewer for their thoughtful comment regarding our choice of a flat grooved structure over a circular grooved geometry. We appreciate the opportunity to clarify our reasoning as follows: The morphology of Pinus sylvestris var. mongolica (PS) needles served as the primary inspiration for our design. Upon detailed examination, it becomes evident that the cross-sectional shape of PS needles is not perfectly circular but instead exhibits a flattened, slightly grooved profile. In light of this observation, our primary objective during the modulation of the nanofiber structure was to replicate this flat grooved morphology, which we successfully achieved with the P1G5 fiber. Consequently, the aerogels were prepared based on the P1G5 fiber structure, ensuring alignment with the natural design principles observed in PS needles. We also acknowledge that the current study
does not include a direct comparison between the properties of cylindrical grooved fibers and flat ribbon-like fibers. This is indeed a valuable direction for future research, and we plan to investigate the comparative performance of these structures in subsequent studies. Thank you for raising this important point, which will undoubtedly help us refine and expand our work in meaningful ways.
Point 6: How did the authors determine the actual level of doping with nano-particles for the U-PG membranes. Clearly there must have been some loss of nano-particles. I find it difficult to believe that all the intentioned nano-particles were attached to the fiber surface.
Response 6: We sincerely thank the reviewer for raising this important question regarding the actual doping level of UiO-66-NH2 in the aerogels. The observed discrepancy between the theoretical and actual doping rates can be attributed to mass loss during the freeze-drying process, a critical step in aerogel fabrication. Freeze-drying effectively removes residual solvents and water while preserving the porous structure; however, it also induces some degree of mass loss, particularly affecting materials with lower densities. In our study, P1G5, being less dense, is more prone to mass loss during freeze-drying. In contrast, UiO-66-NH2, with its significantly higher density, exhibits greater resistance to detachment or removal. Consequently, the relative proportion of UiO-66-NH2 in the final aerogel increases, leading to an actual doping rate that exceeds the theoretical value. This phenomenon is clearly reflected in Table S2, where both the theoretical and actual doping rates are provided, illustrating the impact of the freeze-drying process on the final composition. Further discussion of this effect can be found in lines 302–308 of the revised manuscript. We appreciate the reviewer’s insightful comment.
Point 7: The instrument info and method for mechanical characterization is missing.
Response 7: We thank the reviewer for highlighting the need to include additional details regarding the instrument and method used for mechanical characterization. In response, we have updated the manuscript to provide this information, ensuring greater clarity and reproducibility. The relevant details have been incorporated in lines 134, 140, and 141 of the revised manuscript. We appreciate the reviewer’s feedback, which has helped enhance the quality of our work.

Reviewer 2 Report
Comments and Suggestions for Authors
This study presents a promising approach to air purification using a biomimetic aerogel inspired by the dust-retention properties of Pinus sylvestris var. mongolica needles. The authors cleverly combine electrospinning and freeze-forming technologies to fabricate a grooved ribbon fiber aerogel, further enhanced by the incorporation of UiO-66-NHâ‚‚ nanoparticles for ammonia adsorption. The performance of the UPG-10 aerogel is particularly impressive, exhibiting high filtration efficiency (99.24%), low pressure drop (95 Pa), substantial dust holding capacity (147 g/m²), and significant NH₃ adsorption capacity (99.89 cm³/g). The material also demonstrates good elasticity, retaining over 80% of its original shape after repeated compression cycles.
However, the manuscript would benefit from greater clarity and detail in several areas. Crucially, the authors fail to define "UiO-66-NH₂." Clearly identifying this material is essential for readers to fully understand the composition and functionality. A more detailed explanation of its role in NH₃ adsorption would strengthen the paper.
Furthermore, while the designations PG, UPG-5, UPG-10, UPG-15, and UPG-20 are used to distinguish different aerogel formulations, their precise compositions remain unclear. The authors should provide a table summarizing the composition and relevant physicochemical properties of each aerogel, including the concentrations of incorporated nanoparticles and any other additives. This will greatly improve the reproducibility of the study and facilitate comparison with other materials.
While the dust-retention inspiration from pine needles is mentioned, a more in-depth discussion of the biomimetic principles would enhance the paper’s impact. Specifically, how does the grooved ribbon structure mimic the needle morphology, and how does this contribute to the observed performance improvements? Microscopic images directly comparing the aerogel structure to the needle surface would further support this biomimetic claim. Finally, while the elasticity of the aerogel is a notable advantage, further investigation into its long-term stability and performance under realistic operating conditions would be valuable for assessing its practical applicability.
Author Response
Comments 2:
This study presents a promising approach to air purification using a biomimetic aerogel inspired by the dust-retention properties of Pinus sylvestris var. mongolica needles. The authors cleverly combine electrospinning and freeze-forming technologies to fabricate a grooved ribbon fiber aerogel, further enhanced by the incorporation of UiO-66-NH2 nanoparticles for ammonia adsorption. The performance of the UPG-10 aerogel is particularly impressive, exhibiting high filtration efficiency (99.24%), low pressure drop (95 Pa), substantial dust holding capacity (147 g/m²), and significant NH₃ adsorption capacity (99.89 cm³/g). The material also demonstrates good elasticity, retaining over 80% of its original shape after repeated compression cycles.
However, the manuscript would benefit from greater clarity and detail in several areas. Crucially, the authors fail to define " UiO-66-NH2." Clearly identifying this material is essential for readers to fully understand the composition and functionality. A more detailed explanation of its role in NH₃ adsorption would strengthen the paper.
Furthermore, while the designations PG, UPG-5, UPG-10, UPG-15, and UPG-20 are used to distinguish different aerogel formulations, their precise compositions remain unclear. The authors should provide a table summarizing the composition and relevant physicochemical properties of each aerogel, including the concentrations of incorporated nanoparticles and any other additives. This will greatly improve the reproducibility of the study and facilitate comparison with other materials.
While the dust-retention inspiration from pine needles is mentioned, a more in-depth discussion of the biomimetic principles would enhance the paper’s impact. Specifically, how does the grooved ribbon structure mimic the needle morphology, and how does this contribute to the observed performance improvements? Microscopic images directly comparing the aerogel structure to the needle surface would further support this biomimetic claim. Finally, while the elasticity of the aerogel is a notable advantage, further investigation into its long-term stability and performance under realistic operating conditions would be valuable for assessing its practical applicability.
Response 2: We would like to extend our heartfelt gratitude to the reviewer for taking the time to thoroughly evaluate our manuscript and provide constructive feedback. Your insightful comments and suggestions have greatly contributed to enhancing the quality of our work. We have carefully reviewed each of your points and have made the necessary revisions to address them accordingly. Below, we provide detailed responses to each comment, along with an explanation of the changes we have implemented in the revised manuscript.
Point 1: However, the manuscript would benefit from greater clarity and detail in several areas. Crucially, the authors fail to define "UiO-66-NH2." Clearly identifying this material is essential for readers to fully understand the composition and functionality. A more detailed explanation of its role in NH₃ adsorption would strengthen the paper.
Response 1: We sincerely thank the reviewer for emphasizing the need for greater clarity in defining "UiO-66-NH2" and its role in NH3 adsorption. We fully agree that a clear and comprehensive explanation of this material is essential to help readers better understand its significance within our study. In response to this comment, we have incorporated a more detailed description of UiO-66-NH2 in lines 85–86 of the revised manuscript. Specifically, we now state: "Notably, diverse functionalized variants of UiO-66, especially UiO-66-NH2, exhibit exceptional NH₃ removal performance in various environments [24–26]" in the revised manuscript. This addition underscores the unique properties of UiO-66-NH2 and its
critical contribution to enhancing NH3 adsorption. We believe this clarification significantly improves the overall readability and accessibility of the manuscript. Thank you for your insightful suggestion, which has helped strengthen our presentation.
Point 2: Furthermore, while the designations PG, UPG-5, UPG-10, UPG-15, and UPG-20 are used to distinguish different aerogel formulations, their precise compositions remain unclear. The authors should provide a table summarizing the composition and relevant physicochemical properties of each aerogel, including the concentrations of incorporated nanoparticles and any other additives. This will greatly improve the reproducibility of the study and facilitate comparison with other materials.
Response 2: We sincerely thank the reviewer for their constructive suggestion regarding the need for greater clarity in the compositions of the aerogel formulations (PG, UPG-5, UPG-10, UPG-15, and UPG-20). We fully agree that providing detailed information on the composition and physicochemical properties of each aerogel is essential for improving the reproducibility of our study and facilitating comparisons with other materials. In our study, we have included Table S2 in the Supporting Information, which summarizes the precise composition of each aerogel formulation. This table provides a comprehensive overview of the concentrations of incorporated nanoparticles (e.g., UiO-66-NH2) and any other additives used in the preparation process. By presenting this data in a clear and structured format, we aim to address the reviewer’s concern and enhance the transparency and utility of our work.
Point 3: While the dust-retention inspiration from pine needles is mentioned, a more in-depth discussion of the biomimetic principles would enhance the paper’s impact. Specifically, how does the grooved ribbon structure mimic the needle morphology, and how does this contribute to the observed performance improvements? Microscopic images directly comparing the aerogel structure to the needle surface would further support this biomimetic claim. Finally, while the elasticity of the aerogel is a notable advantage, further investigation into its long-term stability and performance under realistic operating conditions would be valuable for assessing its practical applicability.
Response 3: We sincerely thank the reviewer for their thoughtful and constructive feedback. We fully agree that a more in-depth discussion of the biomimetic principles will significantly enhance the impact of our paper, and we deeply appreciate the opportunity to clarify these aspects. In our team’s previous research, we found that the grooved structure of Pinus sylvestris var. mongolica (PS) needles plays a crucial role in particles interception. In this study, we examined SEM images of the needle surface (Figure 2b) and observed its characteristic micro-grooved structure. To mimic this morphology, we carefully modulated the ratio of PLA to gelatin in the electrospinning solution, successfully preparing the grooved ribbon-like structure in our aerogels. We also compared the filtration performance of UPG-10 aerogels (with grooved ribbon fibers) to conventional PLA and gelatin aerogels, demonstrating superior performance for the biomimetic structure (as shown in Figure 5c). This comparison highlights the functional advantages of the grooved ribbon design in enhancing dust retention and filtration efficiency. Finally, we fully agree with the reviewer on the importance of investigating the long-term stability and performance of the aerogels under realistic operating conditions. While our current study demonstrates the elasticity and initial performance advantages of the aerogels, assessing their durability over extended periods is critical for
practical applications. We plan to address this in future work by conducting rigorous testing under simulated real-world conditions to further optimize their performance and evaluate their practical applicability.
Once again, we are truly grateful for the reviewer’s valuable input, which has significantly helped us improve the clarity, depth, and scientific rigor of the manuscript.
